# A Pan-Transcriptome Analysis Indicates Efficient Downregulation of the FIB Genes Plays a Critical Role in the Response of Alfalfa to Cold Stress

**DOI:** 10.3390/plants11223148

**Published:** 2022-11-17

**Authors:** Xueqi Zhang, Huanhuan Yang, Manman Li, Yan Bai, Chao Chen, Donglin Guo, Changhong Guo, Yongjun Shu

**Affiliations:** Key Laboratory of Molecular Cytogenetics and Genetic Breeding of Heilongjiang Province, College of Life Science and Technology, Harbin Normal University, Harbin 150025, China

**Keywords:** alfalfa, WGCNA, oxidoreductase, ATP synthase, fibrillin

## Abstract

Alfalfa (*Medicago sativa* L.) is a perennial forage legume that is widely distributed throughout the world, and cold stress is an important environmental factor limiting the growth and production of alfalfa in cold regions. However, little is known of the molecular mechanisms regarding cold tolerance in alfalfa. Here, we conducted physiological metabolism assays and pan-transcriptome sequencing on eight cultivars of alfalfa under cold stress conditions. The results of the RNA-seq analysis showed that the genes are “oxidoreductase activity” and “transcription regulator activity”, suggesting that genes with such functions are more likely to play important roles in the response to cold stress by alfalfa. In addition, to identify specific gene modules and hub genes in response to alfalfa cold stress, we applied weighted gene co-expression network (WGCNA) analyses to the RNA-seq data. Our results indicate that the modules of genes that focus on the ATPase complex, ribosome biogenesis, are more likely to be involved in the alfalfa response to cold stress. It is important to note that we identified two fibronectin (FIB) genes as hub genes in alfalfa in response to cold stress and that they negatively regulate alfalfa response to chilling stress, and it is possible that dormant alfalfa is more effective at down-regulating FIB expression and therefore more resistant to cold stress.

## 1. Introduction

Alfalfa (*Medicago sativa* L.) is a high-quality perennial forage legume popularly cultivated worldwide. It not only provides highly nutritious feed for livestock but also benefits environmental systems through symbiotic nitrogen fixation like other leguminous crops [1]. Alfalfa varieties can be simply classified as dormant and non-dormant, based on whether they slow their vegetative growth and store more carbohydrate in the taproot in the fall when daylight hours get shorter, a growth characteristic known as fall dormancy (FD) [1,2]. To facilitate the differentiation of fall dormancy in different alfalfa varieties, fall dormancy classes were assigned to each alfalfa variety ranging from 1 to 11 based on fall growth [1,3]. In addition, based on the FD rating scores, alfalfa varieties can be subdivided into three groups: fall dormant type (FDT, 1–4) with significantly slower fall growth; non-dormant type (NDT, 8–11) with little or no slowdown in fall growth; and semi-dormant type (SDT, 4–7), in between [1,3]. Fall dormancy classification provides an important reference for growing alfalfa and fall dormancy class is often the first trait to be considered when deciding which variety to plant.

Cold tolerance is another important trait of alfalfa to be considered when selecting an alfalfa variety for cultivation, especially in cold regions. Over the past few decades, several studies have found a high correlation between fall dormancy and cold tolerance in alfalfa, with fall dormant alfalfa varieties exhibiting significantly higher winter survival than non-dormant types [4,5,6]. For a long time, there was a misconception that there was a causal relationship between fall dormancy and cold resistance. However, it was gradually recognized that these two traits actually evolved independently in alfalfa as a result of adaptation to different environments [2,3]. Remarkably, Castonguay et al. were able to use indoor screening method with the application of gene-based markers to significantly improve alfalfa freezing tolerance, which is independent of the FD classes of the selected alfalfa varieties, suggesting that genes involved in fall dormancy and cold tolerance are not the same set of genes, although the two traits are closely associated [1]. Nevertheless, the molecular mechanisms regulating these two traits are still poorly understood.

Alfalfa responds to cold stress with the following physiologic metabolic processes: plasma membrane adjustment, the accumulation of cryoprotectant molecules, such as soluble sugar and glycine, as well as the upregulation of the expression of antioxidants, for instance catalase (CAT), peroxidase (POD), and superoxide dismutase (SOD) [1,7,8,9]. However, little is known about the molecular mechanisms which regulate these physiological processes in response to cold stress. RNA-seq has been used extensively in recent years to study the molecular mechanisms of alfalfa development or abiotic stress response, and a fairly large number of key genes have been identified that play critical roles in response to cold stress [3,10,11,12]. All of these studies, however, were restricted to studying only one or two alfalfa cultivars, and as a result, the relevant conclusions obtained can hardly be applied to an overall understanding of the molecular mechanisms of alfalfa cold tolerance.

With the accumulation of gene expression data, the genetic network approach is widely employed to explore gene function or molecular mechanism at system biological level. For example, co-regulatory network can be reconstructed based on gene expression profiles at a series of time points or based on the gene expression profiles of different tissues. Notably, weighted gene co-expression network analysis (WGCNA), currently the most efficient method, has been widely used to identify correlation among genes in studies with large-scale samples [7]. In addition, WGCNA is also helpful to characterize the molecular functions of gene modules, which are composed of co-expressed genes. These co-expressed genes were identified with biological traits through correlation analysis and hub genes conferring to biological traits were effectively identified and characterized [8]. To date, the WGCNA approach has been widely applied to detect co-expressed genes that are responsive to development process in Arabidopsis and Chinese cabbage [9,10], salt stress response in rice [11], and abiotic stress response in Arabidopsis [12,13]. WGCNA has also been used in alfalfa, for example, a series of hub genes involved in the formation of floral pigmentation were determined by transcriptome sequencing combining with WGCNA. In addition, salt and drought stress responsive genes were characterized by the application of WGCNA to RNA-seq data [14,15].

In the present study, to explore the molecular mechanism of fall dormancy and cold tolerance in alfalfa, we focused on eight alfalfa cultivars, four of which are dormant and the other four are non-dormant. We first treated the eight alfalfa cultivars with cold stress, and then tested the physiological metabolism changes in response to cold stress and established and compared their RNA expression profiles after cold stress treatment based on pan-transcriptome sequencing combined with the application of WGCNA. We found that ribosome biogenesis is the key module of co-expressed genes involved in the response to cold stress and that FIBs are the hub and key genes with potential critical roles in the alfalfa response to cold stress.

## 2. Results

### 2.1. Physiological Changes in Alfalfa in Response to Cold Stress

Like other model plants, alfalfa employs a similar set of physiological metabolic processes in response to cold, such as reduced membrane fluidity, the accumulation of cryoprotectant molecules, and the increased expression of antioxidants, such as catalase (CAT) and peroxidase (POD) [1,16,17,18]. However, it is unknown whether there are subtle differences between dormant and non-dormant alfalfa in applying these physiological metabolic processes in response to cold stress. Therefore, we performed tests to compare the activities of POD and CAT and the content of chlorophyl (CHL) and the relative electrolyte conductance (REC) of the eight alfalfa cultivars with/without cold stress treatment. As expected, the tests showed that the POD and CAT activities and REC values in nearly all the cold stress-treated alfalfa cultivars were much higher than the controls, while the CHL contents were lower (Appendix A). However, a two-way ANOVA used to determine the effect of the two variables, FD scores and cold stress, on the above physiological metabolic tests revealed that the eight cultivars had similar levels of POD, CAT, and CHL, but the dormant cultivars showed lower levels of REC compared to non-dormant ones, which were highly changeable when under cold stress treatment (Figure 1). Thus, this suggests that the membrane permeability of dormant alfalfa cultivars is less affected by cold stress compared to non-dormant ones.

### 2.2. Sequencing and De Novo Assembly of Alfalfa Transcriptome

We performed RNAseq for all eight alfalfa cultivars following cold stress treatment with three experimental replicates for each cultivar in order to establish transcriptional profiles in response to cold stress. The constructed libraries were sequenced using next generation sequencing with the total number of 24 samples, and approximately 660 million reads were generated (Appendix A). The cleaned reads were submitted to the NCBI SRA (see PRJNA780579, as previous described) database. To establish a reference alfalfa transcriptome database, we performed the de novo assembly of all alfalfa reads using Trinity and removed redundancy with the CORSET software package (default parameters were used for both software). In the end, we obtained 80,005 transcripts (unigenes) from the de novo assembly with an average length of 1305 bp and an N50 value of 1829 bp (Table 1). All the produced reads in this study were aligned to the reference transcriptome using the software Salmon. The mean mapping rate of individual sample reads to the established reference transcriptome was 79.86%, while the mean mapping rate to the transcriptome of *Medicago truncatula*, alfalfa cv. *Xingjiang Daye* genes, and *ZhongMu No.1* were 75.89%, 72.22%, and 67.19%, respectively (Appendix A), suggesting that our assembled alfalfa transcriptome is more complete for alfalfa comparative transcriptomics or genomics research than the other available ones. The obtained 80,005 alfalfa transcripts were used as queries in a BLAST search against a database established by combining protein sequences from the well-studied plant organisms—*Arabidopsis*, rice, soybean, and *Medicago truncatula*, specifically. We found that approximately 76.8% of transcripts (a total of 61,444 transcripts) have homologues with significant hits (e-value cutoff: 1 × 10^−5^) in the database (Appendix A), which is higher than our previous reports (69.5%) in the alfalfa cv. *Zhaodong*. As expected, longer transcripts had more hits on annotated homologs than short transcripts (Figure 2), implying that longer transcript identities have higher confidence, consistent with previous reports in other species [10].

### 2.3. Functional Annotation Analysis of Alfalfa Transcripts

In order to determine function of alfalfa transcripts, these 80,005 alfalfa transcripts were blasted with model plant proteins. Furthermore, we used the GO terms and KOG annotations of their homologues in the model plant organisms to assign functions to these alfalfa transcripts. For GO term assignment, 36,429 genes (about 45.53%) were grouped into the following GO processes: DNA metabolic process (GO:0006259, 862 unigenes), lipid metabolic process (GO:0006629, 883 unigenes), response to stress (GO:0007267, 120 unigenes), cell–cell signaling (GO:0007267, 203 unigenes), transport (GO:0006810, 116 unigenes), and multicellular organism development (GO:0007275, 160 unigenes) (Figure 3). For KOG annotations, the majority of KOG terms were signal transduction mechanisms (3737 genes in 27,098 KOG annotation hits, 13.79%) (Appendix A). We also identified 2806 plant transcript factors (TF) genes and classified them into 74 TF gene families based on the classification in PlantTFDB. The MYB family (227 genes) was the most abundant member of the TF family, followed by the AP2/ERF (189 genes), WRKY (153 genes), C2H2 (137 genes), and bHLH (135 genes) (Appendix A).

### 2.4. Genes Differentially Expressed in Alfalfa in Response to Cold Stress

In order to identify the differentially expressed genes (DEGs) in the eight alfalfa cultivars under response to cold stress, gene expression levels were compared between the dormant and the non-dormant alfalfa cultivars under cold stress using the DESeq2 package on the R platform. The data showed that compared with the gene expression levels of the non-dormant alfalfa cultivars, 3165 genes of the dormant cultivars are differentially expressed (fold change ≥ 2 or ≤0.5) with 1798 genes were significantly higher expressed and 1367 genes significantly lower expressed (Figure 4). According to the GO annotation analysis of these DEG genes, they are identified with highly enrichment in “oxidoreductase activity” (GO:0016491, 187 genes, adjust *p*-value: 6.5 × 10^−9^), implying oxidoreductase-related genes with critical roles under cold stress and determining cold tolerance in alfalfa (Appendix A). Other molecular functions, such as “acyltransferase activity” (GO:0016747), “tetrapyrrole binding” (GO:0046906), “transcription regulator activity” (GO:0140110), and “metal ion binding” (GO:0046872), were also enriched by a slightly weaker significance, and these functions have been also considered as cold stress response-related in other plants [13].

### 2.5. WGCNA of Alfalfa Response to Cold Stress

We applied WGCNA to our RNA-seq data in order to identify the specific gene modules and the hub genes of alfalfa in response to cold stress. The soft-thresholding power of the adjacency matrix was set to nine based on parameter optimization. Adjacency matrix was calculated, and these genes were clustered and merged into 40 modules. The physiological metabolic assays performed above including CAT, POD, CHL, and REC in both the control and cold stressed groups were used to calculate the correlations with the gene modules. In addition, we normalized the metabolic assay values of the cold stress group to the control, the obtained ratios were used as interaction effect between the two groups, and correlations between gene modules and the ratios were also calculated. In addition, the correlations between the gene modules with the FD scores of the eight alfalfa cultivars were also calculated. For each module, if the correlation with absolute value is larger than 0.5 (>0.5 or <−0.5) and the *p* value is smaller than 0.01, it would be considered as a significant module with effects on either the cold stress responsive physiological metabolism changes or the FD classes of alfalfa. In total, ten, five, four, nine, and seven modules were identified to be significantly correlated with CAT, POD, CHL, REC, and FD, respectively (Figure 5). Among these significant modules, CAT and REC have more common modules with FD, while CHL, and POD have fewer (Appendix A).

Remarkably, the module MElightyellow (cor: 0.8; *p*-value: 3 × 10^−6^) is highly correlated with both FD and REC. This module includes 452 genes that are mainly related to transmembrane transport (GO:0016469, proton-transporting two-sector ATPase complex, *p*-value: 8.5 × 10^−5^, see Appendix A) by GO enrichment analysis (Appendix A). This finding suggests that genes involved in transmembrane transport is upregulated in dormant alfalfa cultivars and therefore cold stress has less effect on their membrane permeability, which is consistent with the above physiological metabolic assay that found smaller changes in REC after cold stress treatment in dormant alfalfa. A slightly less significant module is MEbrown4 (cor: −0.74, *p*-value: 3 × 10^−5^, 326 genes), which contains genes that are mostly related to the ribosome biogenesis process (GO:0042254, *p*-value: 2.1 × 10^−10^) (Appendix A). To generate gene co-expression networks (GCN) with the default parameter, we selected the genes in this module and ultimately obtained 97 genes and 385 connections. Of these 97 genes, two were involved in ribosome biogenesis and are homologous to the fibrillin (FIB) genes of Arabidopsis, which also serve as hub genes in GCN (Figure 6 and Appendix A), implying that the role of FIB genes in response to cold stress is conserved in alfalfa.

## 3. Discussion

### 3.1. Cell Membrane Permeability Plays a Critical Role in Fall Dormant Alfalfa Response to Cold Stress

When the ambient temperature drops to a certain level, alfalfa faces cold stress, which damages alfalfa cell structures and metabolic systems [1]. In response to cold stress, alfalfa adjusts some of its physiological metabolism, including altering cell membrane permeability, elevating electrolytes, activating reactive oxygen species (ROS), and increasing malondialdehyde and POD activity [17]. This whole process is well known as cold acclimation [16,17,19]. Cold acclimation is a common process in plants and is used by a wide variety of plants to resist cold stress, even freezing stress. In this study, we performed a series of assays to determine the physiological metabolism changes of the eight alfalfa cultivars after cold stress treatment. We verified that alfalfa adopts the common cold acclimation in response to cold stress. Specifically, after cold stress treatment, alfalfa showed an increase in MDA and POD, implying the production of ROS, which seriously threatens live plants under cold stress. In present research, alfalfa have been identified as adopting ROS-scavenging method to protect seedlings; as WGCNA analysis shows, oxidoreductase relative genes were characterized with enrichment in three important models (see Figure 6 and Appendix A), which strongly suggested that theROS-scavenging system plays a critical role regarding cold stress. Meanwhile, genes involved in photosynthesis were also identified as enriched in the model MEdarkorange, which is consisted with our physiological analysis with a reduction of chlorophyll levels under cold stress. Moreover, we found that, similar to other plants, the cold acclimation of alfalfa is also regulated by some transcript factors, as the models MEtan and MElightpink4 have shown with numerous TFs genes contained, the molecular fundamentals of which have been extensively studied in model plants, for example, the CBF genes were characterized with key regulation function in alfalfa Zhaodong response to cold stress [20]. However, we detected similar changes in physiological metabolism in all alfalfa cultivars in response to cold stress, independent of the FD scores of the alfalfa cultivars. Given that the FD score is an important indicator of cold tolerance in alfalfa [2,3,6], this finding suggests that alfalfa may have an additional strategy to deal with cold stress that differs between dormant and non-dormant alfalfa cultivars. Through ANOVA analysis, we found that the dormant alfalfa cultivars have significant smaller REC value changes after cold stress relative to the non-dormant ones, which suggests that dormant alfalfa loses less cell membrane permeability after cold stress, which plays a critical role in their resistance to cold stress.

### 3.2. High Expression of ATP Synthase Enhancing Dormant Alfalfa Tolerance to Cold Stress

To characterize functional genes in response to cold stress, we applied WGCNA to the pan-transcriptome sequencing data of the eight cold stress treated alfalfa cultivars and obtained forty modules with co-expressed genes. Interestingly, the most significant module (MElightyellow) is associated with REC. The genes of this module are enriched in the GO term “proton-transporting two-sector ATPase complex” (GO:0033178, five members, *p*-value: 4.9 × 10^−5^), suggesting that in alfalfa ATP, synthase plays important roles in response to cold stress, similar to that in Arabidopsis [21,22]. According to our RNA seq data, the expression of the ATP synthase genes is significantly higher in the dormant alfalfa than in the non-dormant ones after cold stress treatment (Appendix A). This suggests that dormant alfalfa might increase ATP production to provide more energy to deal with cold stress, while the non-dormant cultivars do not have this ability, similar to rice [23] and cucumber [24]. Moreover, there are two phosphatase genes (MsaG27642 and MsaG68441) in this module, suggesting that phosphorylation might play an important role in the molecular mechanism of alfalfa response to cold stress.

### 3.3. Low Expression of FIB Genes Inhibits Alfalfa Growth in Response to Cold Stress

Plant fibrillins (FIBs) are lipid-associated proteins, which are highly conserved from cyanobacteria to higher plants [25,26,27,28]. FIB genes play critical roles in plant growth, development, and response to abiotic and biotic stresses, especially to cold stress [26,28]. The knockdown of FIB1 and FIB2 with RNAi showed that the FIB-deficient lines reduced growth under low temperature stress compared with the parental [28]. In this study, we identified a module, MEbrown4, in which ten genes are annotated as “ribosome biogenesis” (GO:0042254, *p*-value: 2.1 × 10^−10^). Further analysis revealed that two of these ten genes are homologous to FIB1 and FIB2 of Arabidopsis. To determine whether the FIB genes play central roles in this module, we reconstructed the co-expression network by WGCNA, which showed that MsaG03248 (homologous to AtFIB2) is highly connected with the other genes in this module and is referred as a hub-gene. Therefore, similar to Arabidopsis, FIB genes play central roles in the molecular mechanism of alfalfa response to cold stress. Interestingly, the expression level of FIB genes is much lower in the four dormant alfalfa cultivars compared to the four non-dormant alfalfa varieties in our study, indicating that they are negative regulators in response to cold stress. Previous studies have shown that the expression of the FIB genes is lower in dormant alfalfa cultivars than in non-dormant ones under natural outdoor growth conditions and that the expression of the FIB gene is down-regulated in both types of alfalfa in the fall season, see Figure 7 [29]. Taken together, it suggests that FIB genes negatively regulate alfalfa response to cold stress and that the dormant alfalfa is more effective in shutting down the FIB involved pathways and is therefore more resistant to cold stress.

## 4. Materials and Methods

### 4.1. Plant Materials

The seeds of the eight alfalfa (*M. sativa*) cultivars, including Algonquin, WL168HQ, WL319HQ, Bara310SC, WL440HQ, Eureka+, WL525HQ, and WL903HQ, were purchased from Barenbrug China Co., Ltd. (Beijing, China), and their fall dormancy scores are listed in Appendix A.

### 4.2. Alfalfa Growth and Cold Treatment

All seeds of alfalfa cultivars were selected and germinated on filter paper and then transferred to pots, with a mixture of perlite (three volumes) and sand (one volume), as described previously [20]. The seedlings were grown in a chamber and were irrigated with half strength of Hoagland solution every two days. The conditions of chamber were set as 14 h light conditions at 24 °C (as day) and 10 h dark at 18 °C (as night). Alfalfa plants from each cultivar were randomly divided into two groups after eight weeks of growth, one group was set as the control group and maintained in the original chamber, and the other group was set as the cold stress group and transferred to a new chamber at 4 °C. Three hours later, all seedings from both groups were collected and were subjected to physiological analysis and measurements, including peroxidase activity (POD), catalase activity (CAT), total chlorophyll content (CHL), and relative electrolyte conductance (REC). For each cultivar, we collected three biological samples from the control group and three samples from cold stress groups, with five seedlings per biological sample. Each biological sample was quickly frozen in liquid nitrogen and then stored at −80 °C.

### 4.3. Peroxidase Activity Measurements

As described previously [30,31], for each cultivar, we collected three samples from the control group and three samples from cold stress groups, and each sample was leaf tissue collected from five individual seedlings. For each sample, 100 mg fresh leaf tissue was ground with 3 mL of 20 mM KH_2_PO_4_ solution and 0.1 mL protein extraction was mixed with POD solution (0.4 mL 20 mM guaiacol, 0.5 mL 40 mM H_2_O_2_, and 2 mL 50 mM sodium phosphate buffer). The mixture was subjected to absorbance measurement at 470 nm for a continuous period of two minutes, with readings at 15 s intervals. The absorbance differences were used to calculate peroxidase activity based on the formula in previous protocols [30,31].

### 4.4. Catalase Activity Measurements

Catalase activity was measured as previously described [30,31]. Briefly, for each cultivar, we collected three samples from the control group and three samples from cold stress groups, each sample was leaf tissue collected from five individual seedlings. For each sample, 100 mg fresh leaf tissue was ground with phosphate buffer and the protein extraction was mixed with 3 mL H_2_O_2_ solution (30%, *v*/*v*). The mixture was subjected to absorbance measurement at 240 nm for a continuous period of 5 min, with readings at 30 s intervals. The absorbance differences were used to calculate catalase activity based on formula in previous protocols [30,31].

### 4.5. Chlorophyll Contents (CHL) Measurements

As previously described, we collected three samples from the control group and three samples from cold stress groups for each cultivar, each sample were collected leaf tissue from five individual seedlings. For each sample, 100 mg of fresh leaves was mixed with 1 mL of acetone (80%, *v*/*v*) and was ground. After centrifugation for 10 min at 5000× *g*, the supernatant was used to measure the absorbance at 663 and 645 nm. Then the chlorophyll content was calculated as previous protocols described [30,32].

### 4.6. Relative Electrolyte Conductance (REC) Measurements

Relative electrolyte conductance, an index of membrane permeability, was used to evaluate the stability of the cell membrane, which was measured as described in the previous protocol [30,32]. In brief, we collected three samples from the control group and three samples from cold stress groups for each cultivar, each sample was leaf tissue collected from five individual seedlings. For each sample, 200 mg fresh leaves were incubated in 20 mL of distilled water at 25 °C, and EC1 was measured with the incubated leaves at 25 °C for 30 min. Then the incubation was transferred into boiling water for 15 min and EC2 was measured at 25 °C for 15 min. At last, relative electrolyte conductance (shorten as REC) was calculated with the following formula:EC = (EC1/EC2)∗100%.

### 4.7. Construction and Sequencing of RNA-Seq Library

For RNA-seq experiment, we collected three samples from cold stress groups for each cultivar, each sample was leaf tissue collected from five individual seedlings. A total of 200 mg of fresh leaf material was used for total RNA extraction for each sample using the RNeasy Plant Mini Kit (Qiagen, Valencia, CA, USA) according to the manufacturer’s instructions, as described in our previous study. Then, these RNA samples were shipped to BGI-Shenzhen Co., Ltd. (Shenzhen, China). Libraries were constructed according to the manufacturer’s instructions with the mRNA library preparation kit (MGI, Shenzhen, China). RNA-seq was performed on the BGI-Seq platform with the BGI-Seq500 model and 150-bp paired-end reads were generated [33].

### 4.8. Sequence Assembly and Annotation of RNA-Seq Data

First, we removed all the adapter sequences and low qualitative reads from the raw sequence data, then submitted the cleaned reads of these samples to NCBI SRA database (see PRJNA780579, https://dataview.ncbi.nlm.nih.gov/object/PRJNA780579?reviewer=66cer16ae7kbmpq9pahl3ukhmq, visit time: 16 November 2021). Second, we de novo assembled the clean reads into contigs using Trinity with the default parameters [34] and clustered the contigs into unigenes using CORSET with the default parameters as described [35]. All the RNA-seq reads were mapped to four datasets, including our alfalfa unigenes (present research), *Medicago truncatula* transcripts, CDS sequences from two published alfalfa genomes (including cultivars: *Xingjiang Daye* and *Zhongmu No.1*) [36,37], all mapping works were performed using Salmon as a previously described protocol [38]. BLAST searches of these alfalfa unigenes were then performed using BLASTX program for functional annotation (e-value cutoff: 1 × 10^−5^) against the combined database of protein sequences from four model plants, including *Arabidopsis thaliana*, rice (*Oryza sativa*), soybean (*Glycine max*), and *M. truncatula* [39]. We then functionally annotated the alfalfa unigenes based on relevant homologues from the extensive database, including gene ontology (GO) and KOG (EuKaryotic Orthologous Groups) annotations [40,41]. Moreover, the alfalfa unigenes were also scanned in order to identify transcription factors using the iTAK pipeline as described previously [42].

### 4.9. Expression Analysis of Genes Involved in Alfalfa Response to Cold Stress

All clean reads of the RNA-seq were mapped to alfalfa unigenes and the levels of gene expression were quantified using software Salmon quant [38]. The matrix containing the read count mapped to alfalfa unigenes was imported into R platform, and differentially expressed genes were identified using the DESeq2 package, with the parameters set as: the adjustment of *p*-value setting less than 0.01 and the absolute value of fold change value greater than 2, which is manipulated as previously research described [43]. The ggplot2 package of R platform was used for clustering and plotting DEGs, while the GO annotations of these DEGs were retrieved based on the result of previous homologous searches, and the topGO package of R platform was used for GO enrichment analysis [44].

### 4.10. Weighted Gene Co-Expression Network Analysis

The WGCNA package of R platform was used to identify the modules of highly correlated genes based on the basis of expression levels produced by Salmon quant, as previously described [7,38]. The alfalfa unigenes were first filtered on the basis of expression and variance using the “mad” function in the R platform, and 15,000 unigenes were selected with high variation in expression among these alfalfa cultivars in response to cold stress. The soft threshold value 9 was then selected with the “pickSoftThreshold” function of WGCNA package. The adjacency matrix was then generated, and the topological overlap matrix (TOM) was generated using the TOM similarity algorithm. The hierarchical clustering of alfalfa unigenes into modules was performed. We estimated the module-traits associations, including POD, CAT, CHL, REC, and FD, and identified and selected highly correlated modules. We extracted the GO annotations of genes in high correlation modules and performed functional enrichment analysis using the topGO package. At the same time, the selected modules were visualized and analyzed for gene co-expressional network using the Cytoscape software, and hub genes or key genes in co-expressional network were selected for further analysis [45].

## 5. Conclusions

In the present study, in order to investigate the molecular mechanism of alfalfa with different fall dormancy classes in response to cold stress, we performed pan-transcriptome sequencing on eight alfalfa cultivars with cold stress treatment. There were 3165 genes identified as differentially expressed between dormant and non-dormant alfalfa cultivars under cold stress, which were mainly enriched in oxidoreductase activity, transcription regulator activity. The application of WGCNA to the transcriptome data revealed forty modules with co-expression genes, including oxidoreductase genes, ATP synthase genes, and FIB genes. The low expression of FIB genes depresses alfalfa shoot growth under cold stress, which is helpful for maintaining resistance to cold stress.

## Figures and Tables

**Figure 1 plants-11-03148-f001:**
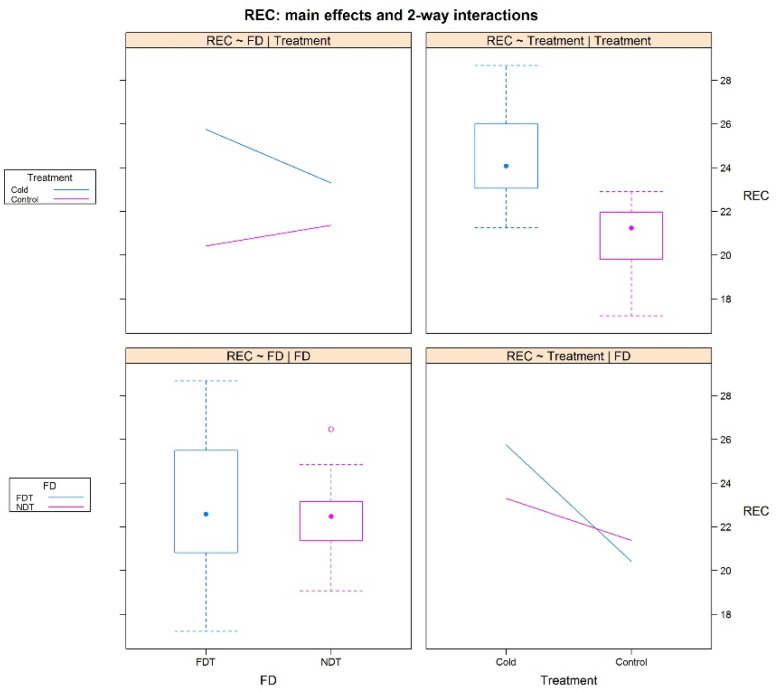
Summary of two-way ANOVA results of alfalfa REC based on FD and cold stress. Note: The ANOVA results shows that cold stress has a significant effect on REC, while FD is not. However, FD and cold stress have an interactive effect on REC in alfalfa, FDT alfalfa have lower REC levels than NDT ones with normal condition, while FDT types present higher REC levels under cold stress. The *X*-axis represents groups, including the FD group and Treatment group, while the *Y*-axis represents the REC value.

**Figure 2 plants-11-03148-f002:**
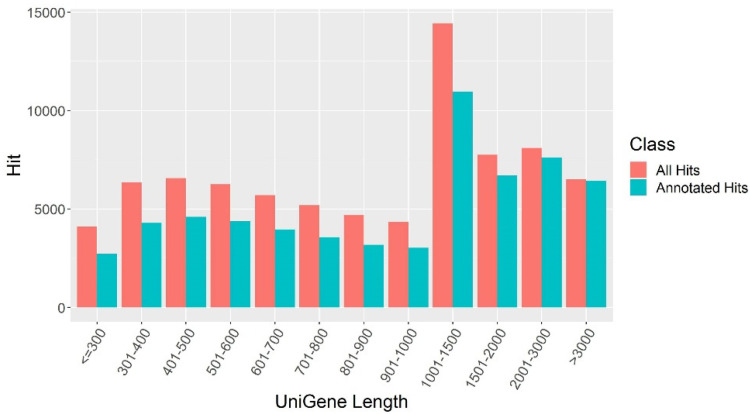
Length distribution of unique transcripts in alfalfa. Note: The *X*-axis lists the length of alfalfa transcripts, and the *Y*-axis lists the numbers of alfalfa transcripts. The red bars represent the total number of alfalfa transcripts in a specific length range, and the adjacent blue bars represent those annotated with model plant genomes.

**Figure 3 plants-11-03148-f003:**
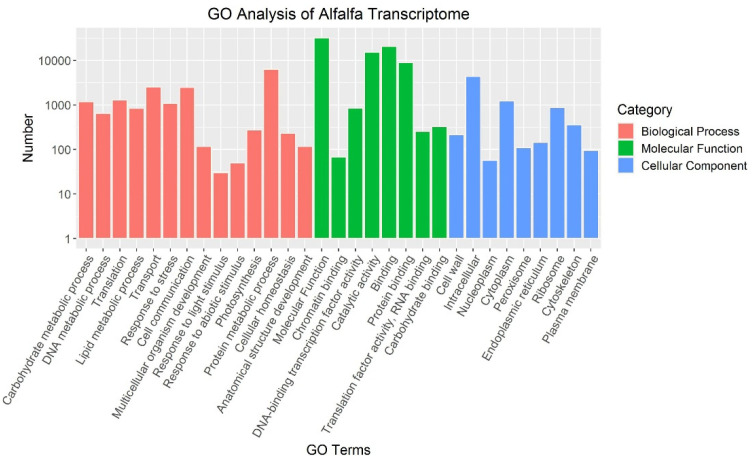
GO annotation of unique alfalfa transcripts. Note: GO terms are listed on the *X*-axis, and the numbers of transcripts grouped in the GO terms are indicated on the *Y*-axis. Red bars represent biological processes; the green bars represent molecular functions; and the blue bars represent cellular components.

**Figure 4 plants-11-03148-f004:**
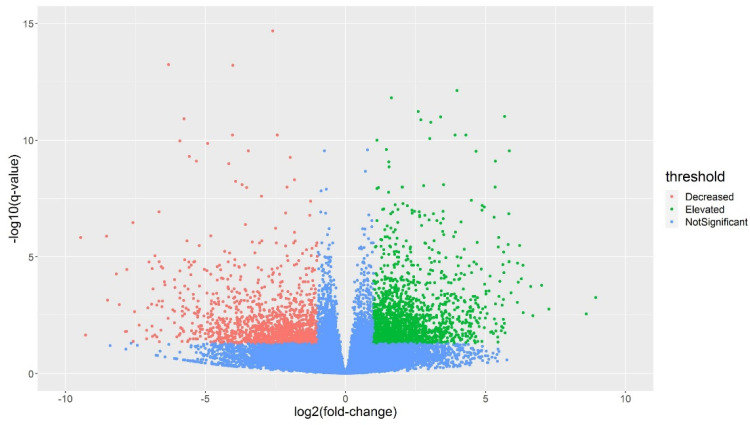
Volcano plots of differentially expressed genes with statistically significant regulation between FDT and NDT groups. Note: Green dots represent genes that are significantly higher expressed in FDT group, while red dots represent genes that are significantly lower expressed in FDT group and blue dots represent those with no significant difference.

**Figure 5 plants-11-03148-f005:**
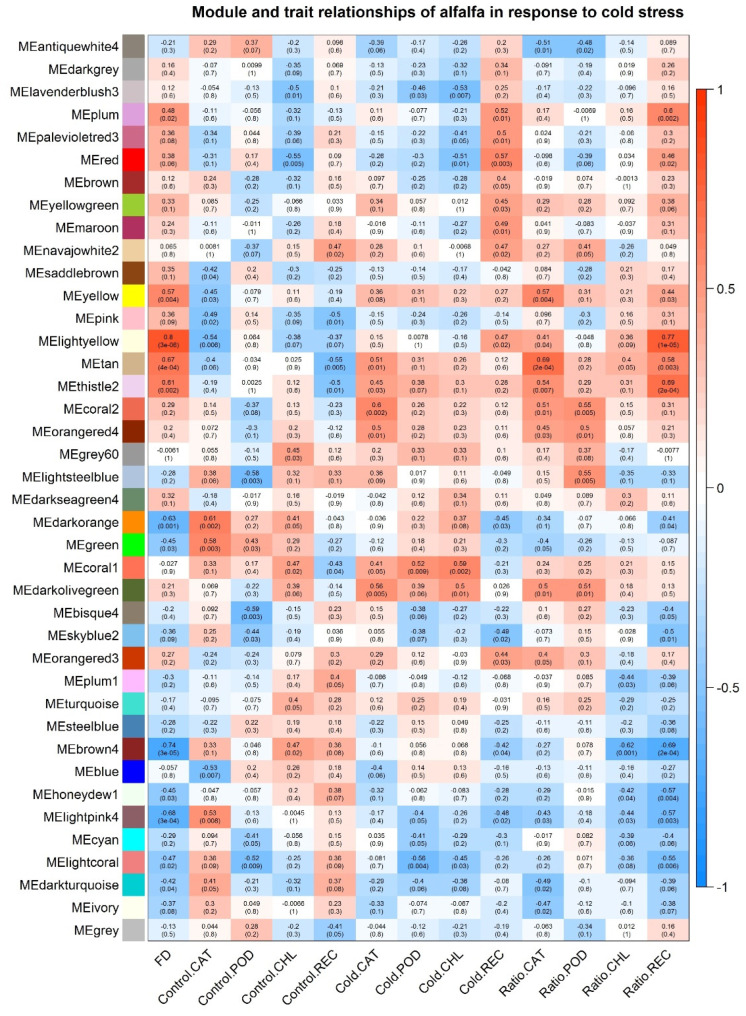
Module-trait associations with the application of WGCNA. Note: Each row corresponds to a module eigengene from their expressional levels and each column to a trait, including FD and physiological metabolic assays. Each cell contains the Pearson correlations and *p*-value of statistical analysis concerning correlations, and the cells were colored based on the Pearson correlations. Red represents a positive coefficient, while blue represents a negative coefficient. FD: fall dormancy; Control: control group; Cold: the group treated with cold stress; and Ratio: the value of the cold stress treated group normalized to that of the control group.

**Figure 6 plants-11-03148-f006:**
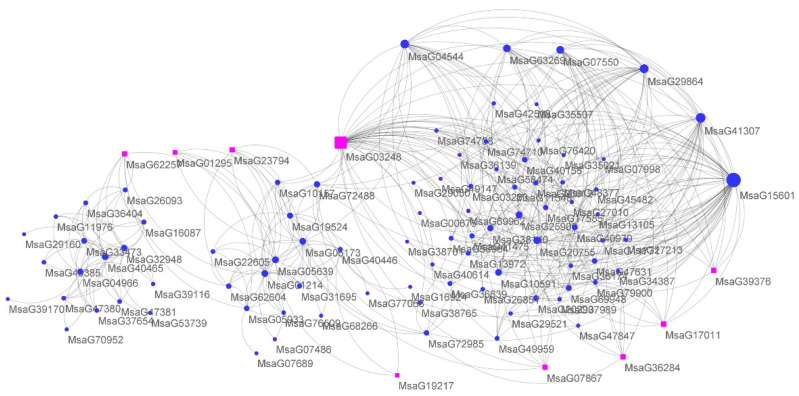
Gene networks in response to cold stress from the model MEbrown4. Note: The modules of genes from model MEbrown4 were exported, and the results and GRN were reconstructed based on the WGCNA results, which were analyzed and displayed using Cytoscape. Gene node size is proportional to the number of interactions (degree), and rectangle shapes correspond to genes involved in ribosome biogenesis process.

**Figure 7 plants-11-03148-f007:**
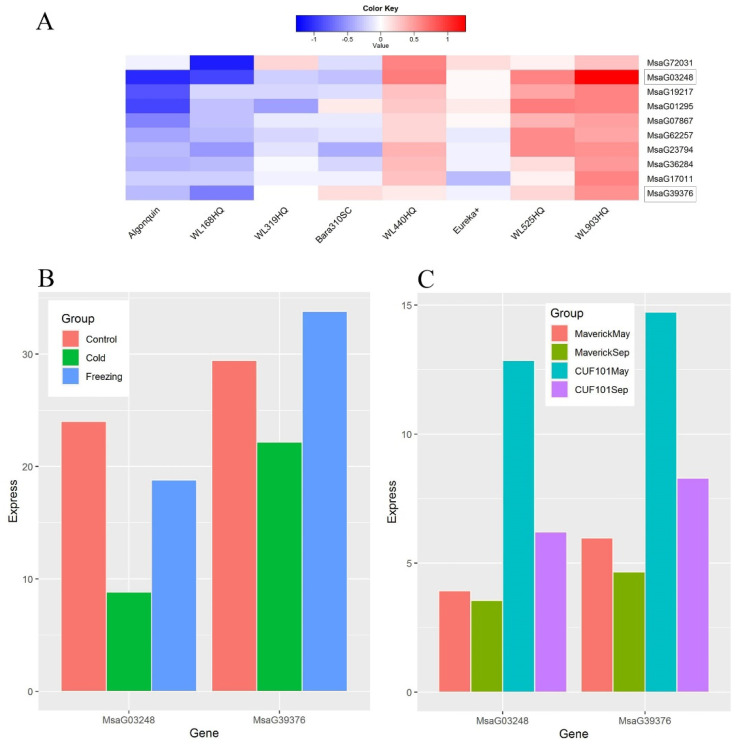
Alfalfa FIB genes expression in response to cold and/or fall dormancy process. Note: (**A**). The expression profiles of alfalfa genes involving in ribosome biogenesis process under cold stress. The FIB genes were marked with rectangle frame, and the RNA-seq data were collected in present research. (**B**). The expression profiles of two alfalfa FIB genes in response to cold and freezing stress, and the RNA-seq data were collected from our previous research [20]. (**C**). The expression profiles of two FIB genes in two cultivars during the fall dormancy process and the RNA-seq data were collected from the research by Zhang et al. [29].

**Table 1 plants-11-03148-t001:** Summary of de novo assembled alfalfa transcriptome.

Data Type	Number
Total sequence	80,005
Number of sequences in 201–500 bp	17,035
Number of sequences in 500–1000 bp	26,180
Number of sequences more than 1000 bp	36,790
Minimal length (bp)	201
Maximal length (bp)	46,546
N50 (bp)	1829
Average length (bp)	1305

## Data Availability

All RNA-seq reads of eight alfalfa cultivars were submitted to the NCBI SRA database, which can be viewed with limitations but can be freely downloaded once the research is published (PRJNA780579, https://dataview.ncbi.nlm.nih.gov/object/PRJNA780579?reviewer=66cer16ae7kbmpq9pahl3ukhmq, accessed on 16 November 2021).

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
