# Peer review of "A Pan-Transcriptome Analysis Indicates Efficient Downregulation of the FIB Genes Plays a Critical Role in the Response of Alfalfa to Cold Stress"

_plants, 2022, doi:10.3390/plants11223148_

Round 1

Reviewer 1 Report

The manuscript “A pan-treanscriptome analysis indicates efficient down-regulation of the FIB genes plays a critical role in the response of alfalfa to cold stress” falls within the journal scope. However, several points need to be pushed further.

1. The pan-treanscriptome data was not validated, such as the expression level of FIB genes by QRT-PCR?

2. L109-110. “the dormant cultivars showed lower levels of REC change when under cold stress treatment compared to the non-dormant ones”. In figure S1, A,B,C,D needed be added, and fig S1D don’t support the above result.

3. In figure 1, lines, charts, X-axis and Y-axis are not well described. The figure legend needs to be supplemented.

4. L102, the expression level of chlorophyl (CHL) or the content of chlorophyl?

5. Reference style is not uniform. For e.g. check authors’ name Dub?, M.-P or Tahir, M.H.;

Author Response

Suggestion: The manuscript “A pan-transcriptome analysis indicates efficient down-regulation of the FIB genes plays a critical role in the response of alfalfa to cold stress” falls within the journal scope. However, several points need to be pushed further.

Answer: Thank you for your positive about our manuscript. The problems about our manuscript were carefully revised, and you will find them point by point as follow.

Q1. The pan-treanscriptome data was not validated, such as the expression level of FIB genes by QRT-PCR?

Answer 1: Thanks for your suggestion. Q-PCR is a golden standard for validating RNA-seq experiment, we should perform qPCR to confirm our results. But, as a result of COVID-19, we are limited to assess our university with working online (Harbin Normal University), and we have difficult to perform qPCR experiment, especially in Harbin. However, RNA-seq results was highly credible, our previous RNA-seq researches have been confirmed, see ShuYJ et al (FPS, 2015), ShuYJ et al (G3, 2016), and Li Wei et al (GMB, 2019). Therefore, we believed our RNA-seq results, in addition, we have collected other alfalfa response to cold stress data, and have confirmed FIB gene expressional profiles (ZhangS et al, PLoS One, 2015 and ShuYJ et al, G3, 2016), see Figure 7 and Figure S9. Thanks for your suggestion.

Shu Yongjun*, Liu Ying, Zhang Jun, Song Lili, Guo Changhong. Genome-wide analysis of the AP2/ERF superfamily genes and their responses to abiotic stress in Medicago truncatula. Front Plant Sci, 2015, 6: 1247.

Shu Yongjun*, Liu Ying, Li Wei, Song Lili, Zhang Jun, Guo Changhong. Genome-wide investigation of microRNAs and their targets in response to freezing stress in Medicago sativa L, based on high throughput sequencing. G3: Genes, Genomes, Genetics, 2016, 6(3): 755-765.

Li Wei, Liu Ying, Zhao Jinyue, Zhen Xin, Guo Changhong, Shu Yongjun*. Genome-wide identification and characterization of R2R3-MYB genes in Medicago truncatula. Genetics and Molecular Biology, 2019, 42(3): 611-623.

Zhang, S.; Shi, Y.; Cheng, N.; Du, H.; Fan, W., and Wang, C., De novo characterization of fall dormant and nondormant alfalfa (Medicago sativa L.) leaf transcriptome and identification of candidate genes related to fall dormancy. PLoS One, 2015, 10, e0122170.

Q2. L109-110. “the dormant cultivars showed lower levels of REC change when under cold stress treatment compared to the non-dormant ones”. In figure S1, A,B,C,D needed be added, and fig S1D don’t support the above result.

Answer 2: Thank you for your suggestions. We have revised our figure S1 with some words (A, B, C, D) added as you suggested. More, there is some ambiguous meaning in our words, the dormant cultivars showed lower levels of REC compared to non-dormant ones, which were highly changing when under cold stress treatment. We have modified our description in our manuscript to eliminate ambiguous mean, you will find them. Thank you.

Q3. In figure 1, lines, charts, X-axis and Y-axis are not well described. The figure legend needs to be supplemented.

Answer 3: Thanks for your suggestion. We have added some words description of axis and legend in note, you will find them. Thank you.

Q4. L102, the expression level of chlorophyl (CHL) or the content of chlorophyl?

Answer 4: Thanks for your suggestion. It is a mistake, which is “the content of chlorophyl”, not “the expression level of chlorophyl”. Thank you again.

Q5. Reference style is not uniform. For e.g. check authors’ name Dub?, M.-P or Tahir, M.H.;

Answer 5: Thanks for your suggestion. We have carefully checked the references, and made them with uniform. Thank you again.

Reviewer 2 Report

The manuscript seems to be well written, and the results presented logically. However, I have general comments for the authors to consider and improve their manuscript. I think the discussion was too short and Alfalfa adaptation to cold stress was not deeply discussed. Besides the FIB and ATPase expression in response to cold stress, are there other accessory genes/transcripts that could be involved in cold stress tolerance in Alfalfa. If so, then how did these genes interact in the predicted network.  

1. Are there nuclear factors that changed in response to cold stress? i.e., TFs and genes in well-known hormonal pathways involved in stress response that could cause a global shift in transcription that affects downstream factors. 

2. Did the authors see changes in the transcripts of genes coding antioxidant enzymes/proteins as suggested in the physiological data?

3. Did the authors validate the expression of the major transcripts they have proposed as key candidates for cold stress adaptation in Afalfa?  I am not sure figures 7 B and C are qPCR results. I think it is important to validate the expression of these genes at least in the biological replicates of the samples used for RNAseq. 

Author Response

Suggestion: The manuscript seems to be well written, and the results presented logically. However, I have general comments for the authors to consider and improve their manuscript. I think the discussion was too short and Alfalfa adaptation to cold stress was not deeply discussed. Besides the FIB and ATPase expression in response to cold stress, are there other accessory genes/transcripts that could be involved in cold stress tolerance in Alfalfa. If so, then how did these genes interact in the predicted network.

Answer: Thank you for your positive about our manuscript. The problems about our manuscript were carefully revised, and you will find them point by point as follow.

Q1. Are there nuclear factors that changed in response to cold stress? i.e., TFs and genes in well-known hormonal pathways involved in stress response that could cause a global shift in transcription that affects downstream factors.

Answer 1: Thank you for your suggestions. TFs were also present with critical roles response to cold stress in our research. We have displayed more models of WGCNA results, and TFs annotated with GO ID GO:0003700 were enriched in two models MEtan and MElightpink4, as Table S5 shown, we have revised our manuscript with new table. However, TFs may be rapidly responsive to cold stress, for example, they have quickly changed expressional levels at 30 minutes, as our research in white clover (Zhang et al, Biotechnology Biotechnological Equipment, 2022). While their expressional levels with smaller changes than other function genes at three hours point, which was also characterized in white clover. There may be a global shift in transcription as you suggested, which would be validated with more experiments in future. Thank you.

Zhang XQ, et al. Time-course RNA-seq analysis provides an improved understanding of genetic regulation in response to cold stress from white clover (Trifolium repens L.), Biotechnology Biotechnological Equipment, 2022, 29(1): 745–752.

Q2. Did the authors see changes in the transcripts of genes coding antioxidant enzymes/proteins as suggested in the physiological data?

Answer 2: Thank you for your suggestions. Many genes with oxidoreductase activity were enriched in three co-expression models, which were annotated with GO:0016491 (oxidoreductase activity) and GO:0016717 (oxidoreductase activity, acting on paired donors, with oxidation of a pair of donors resulting in the reduction of molecular oxygen to two molecules of water), see Table S5, we have added these results in our revision, and modified results and discussion. Thank you.

Q3. Did the authors validate the expression of the major transcripts they have proposed as key candidates for cold stress adaptation in Afalfa?  I am not sure figures 7 B and C are qPCR results. I think it is important to validate the expression of these genes at least in the biological replicates of the samples used for RNAseq.

Answer 3: Thanks for your suggestion. Q-PCR is a golden standard for validating RNA-seq experiment, we should perform qPCR to confirm our results. But, as a result of COVID-19, we are limited to assess our university with working online (Harbin Normal University), and we have difficult to perform qPCR experiment, especially in Harbin. However, RNA-seq results was highly credible, our previous RNA-seq researches have been confirmed, see ShuYJ et al (FPS, 2015), ShuYJ et al (G3, 2016), and Li Wei et al (GMB, 2019). Therefore, we believed our RNA-seq results, in addition, we have collected other alfalfa response to cold stress data, and have confirmed FIB gene expressional profiles (ZhangS et al, PLoS One, 2015 and ShuYJ et al, G3, 2016), see Figure 7 and Figure S9. Thanks for your suggestion.

Shu Yongjun*, Liu Ying, Zhang Jun, Song Lili, Guo Changhong. Genome-wide analysis of the AP2/ERF superfamily genes and their responses to abiotic stress in Medicago truncatula. Front Plant Sci, 2015, 6: 1247.

Shu Yongjun*, Liu Ying, Li Wei, Song Lili, Zhang Jun, Guo Changhong. Genome-wide investigation of microRNAs and their targets in response to freezing stress in Medicago sativa L, based on high throughput sequencing. G3: Genes, Genomes, Genetics, 2016, 6(3): 755-765.

Li Wei, Liu Ying, Zhao Jinyue, Zhen Xin, Guo Changhong, Shu Yongjun*. Genome-wide identification and characterization of R2R3-MYB genes in Medicago truncatula. Genetics and Molecular Biology, 2019, 42(3): 611-623.

Zhang, S.; Shi, Y.; Cheng, N.; Du, H.; Fan, W., and Wang, C., De novo characterization of fall dormant and nondormant alfalfa (Medicago sativa L.) leaf transcriptome and identification of candidate genes related to fall dormancy. PLoS One, 2015, 10, e0122170.

Round 2

Reviewer 2 Report

The authors have addressed most of the points I raised.